# Non-monotonic pressure dependence of the thermal conductivity of boron arsenide

Navaneetha K. Ravichandran[1] & David Broido[1]

Recent experiments demonstrate that boron arsenide (BAs) is a showcase material to study the role of higher-order four-phonon interactions in affecting heat conduction in semiconductors. Here we use first-principles calculations to identify a phenomenon in BAs and a related material - boron antimonide, that has never been predicted or experimentally observed for any other material: competing responses of three-phonon and four-phonon interactions to pressure rise cause a non-monotonic pressure dependence of thermal conductivity, $\kappa$, which first increases similar to most materials and then decreases. The resulting peak in $\kappa$ shows a strong temperature dependence from rapid strengthening of four-phonon interactions relative to three-phonon processes with temperature. Our results reveal pressure as a knob to tune the interplay between the competing phonon scattering mechanisms in BAs and similar compounds, and provide clear experimental guidelines for observation in a readily accessible measurement regime.

[1] Department of Physics, Boston College, Chestnut Hill, MA 02467, USA. Correspondence and requests for materials should be addressed to N.K.R. (email: navaneeth.ravichandran@bc.edu)

Understanding thermal transport through semiconductors and insulators, where heat is carried by quantized lattice vibrations called phonons[1,2], has been a topic of intense research interest over the past century. Of particular interest is the behavior of the thermal conductivity, $\kappa$, of non-metallic solids under the application of hydrostatic pressure, which has been experimentally investigated for almost 100 years[3]. Over these years, the pressure dependence of $\kappa$ for numerous non-metals including alkali halides[4–6], ice[7,8], and those residing in the Earth's interior, such as magnesium oxide (MgO)[9], magnesium silicate (MgSiO₃)[10], mica[11], and quartz[12–14] has been reported in the literature. In all of these materials, $\kappa$ was found to increase monotonically with pressure except near a structural transition where it could decrease monotonically (e.g., see ref. [15] for $\kappa$ of copper chloride under pressure).

From a theoretical perspective, the observed monotonic pressure dependence of $\kappa$ has been typically described using simple parameterized models[4–14]. Only recently have first-principles calculations been developed that are able to capture the changes in $\kappa$ with pressure quantitatively[16–18], and they have also found that $\kappa$ changes monotonically with pressure for these materials.

The intrinsic thermal resistance in non-metallic crystals arises from the mutual interactions among phonons[1,2]. Typically, it has been assumed that the lowest-order interactions among three phonons are sufficient to predict the thermal conductivity of materials[1,2], and the higher-order phonon–phonon interactions are neglected. This is also the case with the aforementioned ab initio calculations[16–18] for the pressure dependence of $\kappa$. While this three-phonon scattering limited ab initio framework has been demonstrated to work well to describe the thermal conductivity of many compounds[19–25], a recent ab initio calculation[26] has predicted that the semiconductor, boron arsenide (BAs), should have unusually weak three-phonon interactions leading to an unconventionally ultra-high $\kappa$ at ambient pressure, and a subsequent ab initio work[27] predicted that higher-order four-phonon interactions can be of comparable strength to three-phonon processes, thus lowering $\kappa$. These predictions of the important role played by four-phonon scattering in affecting $\kappa$ of BAs have also been confirmed by three recent independent experimental works[28–30]. Thus, it is unclear if the apparent universal monotonic pressure dependence of $\kappa$, supported by measurements on dozens of materials, will hold for BAs.

In this work, we show from first-principles calculations that opposing responses of three-phonon and four-phonon scattering strengths to hydrostatic pressure cause a non-monotonic pressure dependence of $\kappa$ in BAs and a related material, boron antimonide (BSb)—a phenomenon that has never been predicted or observed experimentally for any other material. We show that these opposing responses first increase and then decrease $\kappa$ with increasing pressure, and the resulting $\kappa$-peak position also shows striking dependencies on temperature from rapid strengthening of four-phonon processes. Our results show that the unusual microscopic features of phonons and their mutual interactions in BAs and similar compounds are responsible for the competition that results in the unique pressure and temperature dependencies of $\kappa$, and they shed light on the important role played by higher-order phonon processes in semiconductors and insulators.

## Results

### Pressure-driven changes to thermal conductivity of solids.
To calculate $\kappa$ of a solid as a function of pressure and temperature, we have implemented a predictive, first-principles approach with no adjustable parameters, that has demonstrated good agreement with the measured lattice expansion, temperature-dependent phonon frequencies, and thermal conductivity of diamond and sodium chloride[31], and the ultra-high thermal conductivity of BAs[28], where both three-phonon and four-phonon scattering are important. This recently developed approach goes beyond the standard calculations by including higher-order four-phonon scattering processes along with three-phonon processes in describing phonon transport, thereby providing a new opportunity to investigate the significance of higher-order processes and their interplay with those of lower order on the thermal properties of materials. The details of this unified first-principles computational framework are summarized in the Methods section. These calculations present major computational challenges, and we have developed several efficiencies in our code to overcome them, as described in the Methods section.

In general, application of hydrostatic pressure increases the phonon frequencies and group velocities of a material; hence qualitatively one might expect $\kappa$ to increase with pressure. Figure 1a shows such stiffening of phonon modes for MgO with pressure from our calculations which compare well with those observed experimentally[32–34]. Good agreement with the measured pressure dependence of volume and phonon frequencies is obtained for MgO and cubic boron nitride (cBN) [see Supplementary Note 1]. Our $\kappa$ calculations for MgO and cBN show that $\kappa$ indeed increases monotonically with pressure in Fig. 1b, c, respectively, in-line with the conventional expectation. In particular, Fig. 1b shows that our calculations for $\kappa$ of MgO with natural isotopic mix of the constituent elements is in reasonable quantitative agreement with the experimental data[9] [see Supplementary Note 9 for the effect of point defects on the thermal conductivity of MgO]. Furthermore, the effect of four-phonon scattering is weak in these materials, contributing to only ~5–15% reduction from the three-phonon limited $\kappa$ for MgO and ~1–9% for cBN in the range of pressures considered in this work.

Although there are a few exceptions in the literature reporting observations of monotonically decreasing thermal conductivity of materials with pressure (P) rise ($\frac{\partial \kappa}{\partial P}<0$ in e.g., ref. [15] for copper chloride), these anomalous observations are mostly associated with the proximity of the experimental conditions to structural phase transitions. The commonly used Leibfried–Schlomann (LS) theory has been found to be useful in providing trends to the behavior of $\kappa$ under pressure[6,7,9,35]. According to the LS theory[36],

$$\frac{B_T}{\kappa}\left(\frac{\partial \kappa}{\partial P}\right)_T = g \qquad (1)$$

where $B_T$ is the isothermal bulk modulus ($B_T > 0$) and $g$ is a constant. Note that Eq. (1) admits no other possibility than a monotonically increasing or decreasing $\kappa$ with increasing pressure. However, the LS theory was derived under many approximations, and it is not predictive. In particular, it fails to provide a correct picture for the pressure-dependent $\kappa$ of BAs. Specifically, it predicts that the $\kappa$ of BAs increases monotonically with increasing pressure[18] contrary to our findings below.

### Thermal conductivity of BAs under pressure.
To examine the pressure dependence of $\kappa$ for BAs, we begin by calculating from first principles, $\kappa^{(3)}$, the thermal conductivity of BAs including only three-phonon scattering, as a function of pressure. Figure 2b shows that $\kappa^{(3)}$ decreases monotonically with pressure for several different temperatures between 200 K and 1000 K, contrary to the prediction from LS theory, but consistent with the previously reported first-principles calculations at 300 K[18]. It occurs because of specific features of the phonon dispersions in BAs, shown in Fig. 2a. At ambient pressure, the large frequency gap between acoustic and optic phonons combined with the small bandwidth of optic phonon frequencies prevents three-phonon scattering of the heat carrying acoustic phonons by optic phonons[26].

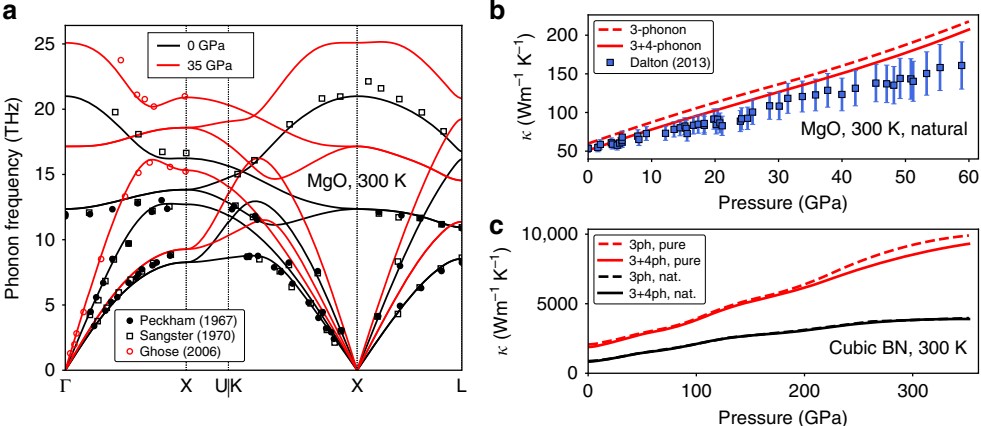

**Fig. 1** Phonons and thermal conductivity of MgO and cBN under pressure. **a** Phonon dispersions for MgO at 300 K computed at 0 GPa (black) and 35 GPa (red) pressures. The calculations are in good agreement with the measurements of phonon frequencies at 0 GPa[32,33] and 35 GPa[34]. **b** Three-phonon and 3 + 4-phonon limited $\kappa$ of MgO versus pressure at 300 K compared with experiments[9]. **c** Three-phonon and 3 + 4-phonon limited $\kappa$ of cBN (red curves: isotopically pure $^{11}B^{14}N$, black curves: naturally occurring cBN) versus pressure at 300 K. Our results for cBN differ from those in ref. [46], where a bimodal behavior of the three-phonon limited $\kappa$ for cBN with increasing pressure was observed, but was later identified[18] to be erroneous and related to required symmetries of the inter-atomic force constants used in ref. [46] not being satisfied

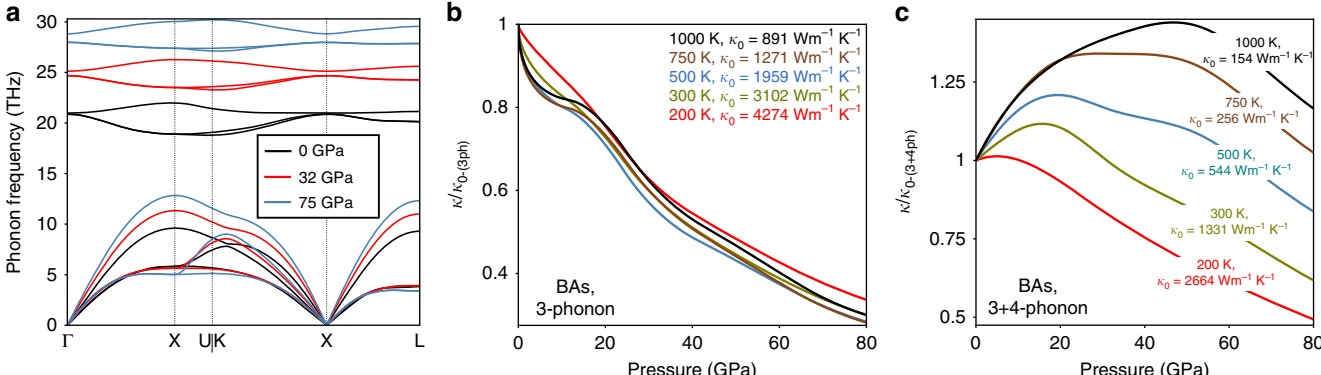

**Fig. 2** Phonons and thermal conductivity of BAs under pressure. **a** Phonon dispersions of BAs at 300 K from 0 GPa to 75 GPa pressure. **b** Three-phonon limited $\kappa$ of BAs versus pressure at different temperatures. **c** 3 + 4-phonon limited $\kappa$ of BAs versus pressure at different temperatures. All $\kappa$ curves are scaled by the corresponding zero-pressure values listed within the figures. BAs shows a rapidly decreasing three-phonon $\kappa$, while the 3 + 4-phonon value shows an unusual temperature-dependent non-monotonic behavior with pressure

Also, three-phonon scattering among acoustic phonons is reduced because phonon frequencies of the three acoustic phonon branches are bunched closer together than in most materials (note e.g., the proximity of the higher transverse acoustic (TA) branch and the longitudinal acoustic (LA) branch along the $\Gamma \rightarrow K$ direction at 0 GPa in Fig. 2a)[26]. With increasing pressure, the optic and the LA phonon branches of BAs show stiffening with pressure in Fig. 2a similar to the phonon dispersions of MgO in Fig. 1a, while the TA branches show weaker pressure dependence. As a result, the three-phonon scattering channels involving acoustic and optic phonons are still forbidden, and the increased separation of LA and TA phonons with pressure decreases the bunching effect[18,26], increases the phase space for three-phonon scattering between acoustic phonons and decreases $\kappa^{(3)}$. Since the zinc-blende structure of BAs is known to be stable to above 100 GPa at room temperature[37], the prediction of $\frac{\partial \kappa^{(3)}}{\partial P} < 0$ even around atmospheric pressure is already a strong deviation from observations in literature for other materials[4–6,9,11], and highlights the importance of carefully accounting for phonon–phonon scattering phase space effects in the theories for $\kappa$.

When four-phonon scattering is included, the 3 + 4-phonon limited thermal conductivity, $\kappa^{(3+4)}$ (Fig. 2c), shows three stark qualitative and quantitative differences from $\kappa^{(3)}$ (Fig. 2b). First, with increasing pressure, the $\kappa^{(3+4)}$ curves initially increase and then decrease, resulting in a $\kappa^{(3+4)}$-peak for each temperature. Second, the position of the $\kappa^{(3+4)}$-peak shifts to higher pressures as the temperature is increased. Third, the percentage enhancement in $\kappa^{(3+4)}$ from the zero-pressure value at the $\kappa^{(3+4)}$-peak is larger at higher temperatures. These three features are unique to BAs (and also to a related material, BSb, as we show later) and have never been experimentally observed or computationally predicted in any other material. The value for $\kappa^{(3+4)}$ at ambient pressure is in good agreement with measured data on high quality BAs crystals[28–30] [similar results were obtained on BAs with naturally occurring isotopic mix of the constituent atoms, see Supplementary Note 2].

**Pressure-driven competition among scattering channels.** The non-monotonic behavior of $\kappa^{(3+4)}$ in BAs is caused by the competing responses of three-phonon and four-phonon scattering rates as the hydrostatic pressure increases, as elucidated in Fig. 3a–d, where the total three-phonon and process-wise classified four-phonon scattering rates versus phonon frequency are compared at different pressures at 300 K. First observe that in

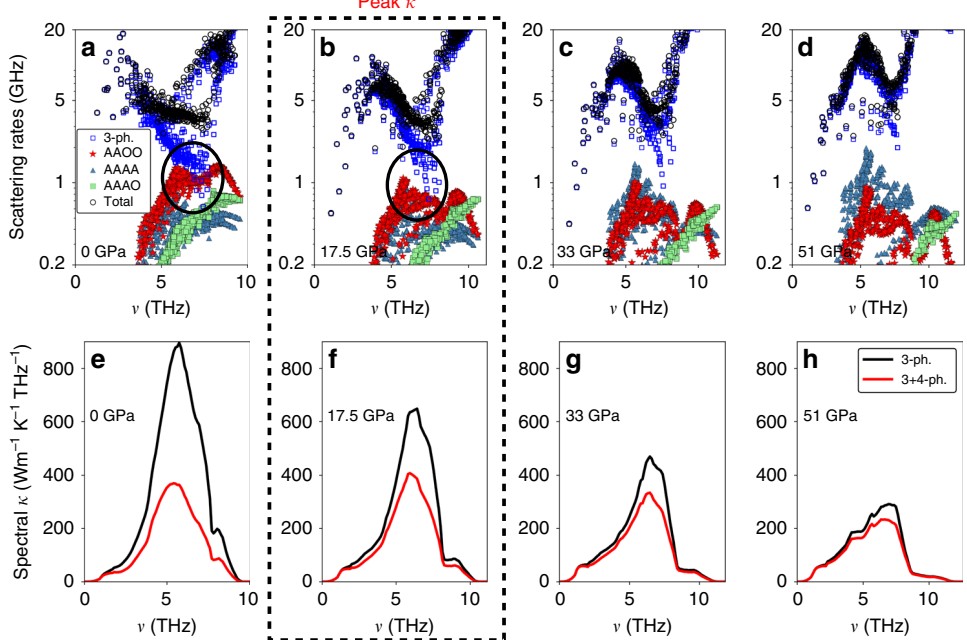

**Fig. 3** Opposing responses of three- and four-phonon scattering strengths to pressure. **a–d** Total three-phonon and process-wise four-phonon scattering rates of the acoustic phonons in BAs as a function of the phonon frequency ($\nu$) at 300 K and at different pressures. As pressure increases, total three-phonon scattering rates and AAAA four-phonon scattering rates strengthen, but AAAO and AAOO four-phonon scattering rates weaken. The competing responses to increasing pressure from three-phonon processes and the dominant AAOO four-phonon processes below 20 GPa cause a thermal conductivity peak at ~17.5 GPa (shown by the dashed box). Beyond 50 GPa, three-phonon scattering completely dominates over four-phonon processes; thus the strengthening AAAA four-phonon scattering rates have a very weak effect on the pressure dependence of $\kappa^{(3+4)}$. The oval regions in **a** and **b** show the frequency range that contributes the most to the $\kappa$ of BAs. **e–h** Spectral contributions to $\kappa^{(3)}$ and $\kappa^{(3+4)}$ as a function of $\nu$ at different pressures for acoustic phonons. The frequency range of 4–8 THz produces the largest contribution to both $\kappa^{(3)}$ and $\kappa^{(3+4)}$

Fig. 3a, the frequency range where the three-phonon scattering rates are the weakest at zero pressure is exactly where the dominant four-phonon scattering rates involving two acoustic and two optic phonons (AAOO), are the strongest. In this frequency region, the three-phonon and AAOO four-phonon scattering rates have comparable strengths. This region contributes the most to the $\kappa^{(3+4)}$ of BAs[28], as shown by the spectral $\kappa^{(3)}$ and $\kappa^{(3+4)}$ in Fig. 3e. In this region, marked by the oval in Fig. 3a, inclusion of four-phonon scattering increases the total phonon–phonon scattering rates significantly, reducing the calculated $\kappa^{(3+4)}$ by about a factor of two at room temperature and ambient pressure, and more at higher temperatures compared to $\kappa^{(3)}$, as elucidated in refs. [27,28] [see Supplementary Note 4 for details on the absolute strength of the four-phonon scattering channels].

As explained above, the three-phonon scattering rates in BAs, which involve almost exclusively scattering among three acoustic phonons (AAA processes)[18,26], increase with pressure [Fig. 3a–d], resulting in a decreasing $\kappa^{(3)}$ with pressure rise. On the other hand, pressure rise has the opposite effect on the AAOO four-phonon scattering processes, which are the dominant four-phonon scattering channels up to ~20 GPa at 300 K. The reduction in AAOO four-phonon scattering rates with pressure rise comes in part from a decreasing scattering phase space and increasing optic phonon frequencies (which causes a reduction in the scattering matrix elements and Bose factors that determine the strength of the scattering channels) with pressure rise [see Supplementary Note 5 for details on the pressure dependence of three-phonon and four-phonon scattering phase space]. Initially, the reduction in AAOO scattering rates has a larger effect on $\kappa^{(3+4)}$ than does the increase in AAA scattering strength with pressure. Therefore, the total scattering rates initially

weaken, resulting in an increasing $\kappa^{(3+4)}$ with pressure. Then, as the lowest three-phonon scattering rates continue to increase with pressure, they eventually dominate the total scattering rates (Fig. 3b–d). Thus, $\kappa^{(3+4)}$ starts decreasing, as in Fig. 2c, which results in a $\kappa^{(3+4)}$-peak for all temperatures at and above 300 K. The opposite responses of three-phonon and four-phonon scattering rates to pressure rise are caused by the unique dispersion of BAs and do not occur for most other materials [e.g., see Supplementary Note 1 for three-phonon and four-phonon scattering rates of MgO and cBN with pressure at 300 K].

**Temperature-driven shift of the thermal conductivity peak.** The second interesting feature of the pressure dependence of $\kappa^{(3+4)}$ in BAs is that the pressure at which $\kappa^{(3+4)}$ peaks, increases with increasing temperature. Figure 4a shows a pressure-temperature contour plot of the distribution of $\kappa^{(3+4)}$ scaled by the value at zero pressure and the same temperature $\left(\kappa_0^{(3+4)}\right)$ along with the changing position of the $\kappa^{(3+4)}$-peak at different temperatures. The root cause of this unusual behavior in BAs is related to the larger four-phonon scattering rates relative to three-phonon scattering at elevated temperatures, as shown in Fig. 4 (b) [also see Supplementary Note 6 for the full temperature dependence of the scattering rates]. Compared to 300 K, the four-phonon scattering rates at zero pressure and 750 K are significantly stronger than the three-phonon scattering rates in the region of maximum spectral contribution to $\kappa^{(3+4)}$ (4 to 8 THz). As a result, when pressure is increased at 750 K, the relevant three-phonon scattering rates do not exceed the four-phonon scattering rates until much higher pressure than is the case at 300 K. This phenomenon also gives rise to the third interesting feature: since the cross-over point between three-phonon and

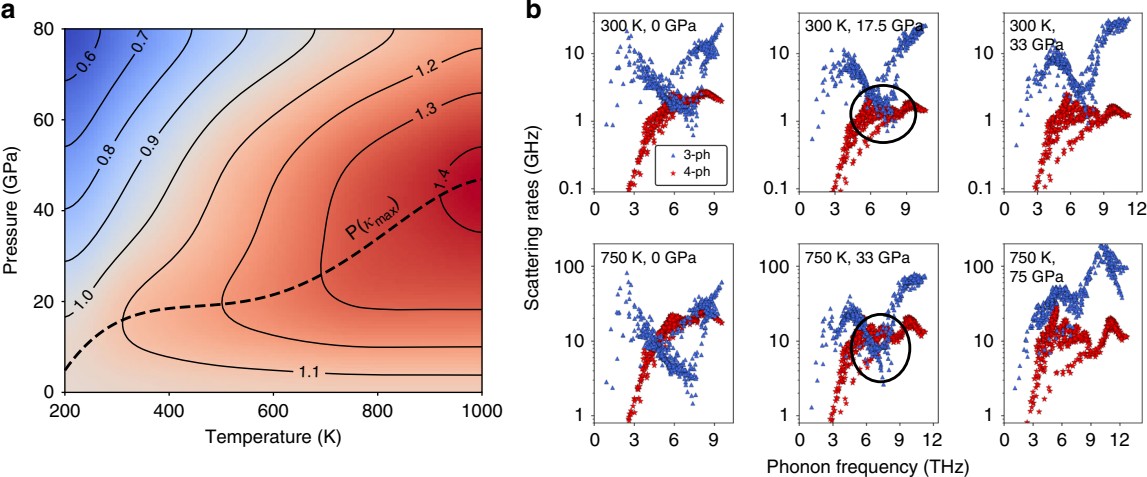

**Fig. 4** Strengthening four-phonon scattering with temperature shifts the $\kappa^{(3+4)}$-peak in BAs. **a** Pressure-temperature colormap of $\kappa^{(3+4)}$ scaled by the corresponding zero-pressure value at each temperature $\left(\kappa_0^{(3+4)}\right)$ along with the iso-$\kappa^{(3+4)}/\kappa_0^{(3+4)}$ lines, showing the non-monotonic pressure dependence of $\kappa^{(3+4)}$ at each temperature beyond 300 K. Also shown is the shifting position of the thermal conductivity peak on the pressure-temperature surface. **b** Comparison of three-phonon and four-phonon scattering rates of the acoustic modes in BAs at 300 K and 750 K at different pressures. The center plots [(300 K, 17.5 GPa) and (750 K, 33 GPa)] are the conditions close to the $\kappa^{(3+4)}$-peak in Fig. 2c at the corresponding temperatures

four-phonon scattering is delayed until higher pressures at higher temperatures, the $\kappa^{(3+4)}$ below the cross-over point keeps increasing with pressure rise. Thus, the percentage enhancement in $\kappa^{(3+4)}$ at the $\kappa^{(3+4)}$-peak from the zero-pressure value is larger at elevated temperatures.

We have also computed $\kappa^{(3+4)}$ for BSb, which was also predicted to possess ultra-high $\kappa^{(3)}$ in ref. [26]. Since BSb has the same features in its phonon dispersions as does BAs, four-phonon scattering is expected to play a role in reducing its thermal conductivity. As shown in the Supplementary Note 7, both isotopically pure BSb and BSb with natural isotopic mix show a strong suppression in $\kappa$ from four-phonon scattering at zero-pressure and show a $\kappa^{(3+4)}$ peak in the pressure dependence (Fig. 5), which is qualitatively similar to that in BAs.

## Discussion

In summary, we have used rigorous first-principles calculations to predict an unusual feature in the thermal conductivity of non-metals, unique to BAs and BSb, that has never been experimentally observed or predicted theoretically: the competing responses of three-phonon and four-phonon scattering to pressure rise results in a non-monotonic pressure dependence of $\kappa$. The resulting peak in $\kappa$ shifts in position to higher pressure with increasing temperature because of the more rapid strengthening of the four-phonon processes compared to three-phonon processes. Furthermore, this difference in the temperature dependence of three-phonon and four-phonon processes also results in a larger percentage enhancement of $\kappa$ from its zero-pressure value, at higher temperatures.

Apart from the ultra-high thermal conductivity of BAs at ambient conditions, existing literature has also predicted that BAs possesses excellent electron and hole mobilities as well[38], and that the application of hydrostatic pressure could enhance its electronic properties further[39]. These studies have also predicted enhanced carrier mobilities in BSb at ambient conditions and under pressure, for the same reasons as in BAs. Our results reveal the unusual way in which pressure tunes the ultra-high thermal conductivity of BAs and BSb, and also emphasize the importance of including higher-order phonon processes to explore new heat transfer regimes that upend conventional understanding. With the recent advances in the growth and

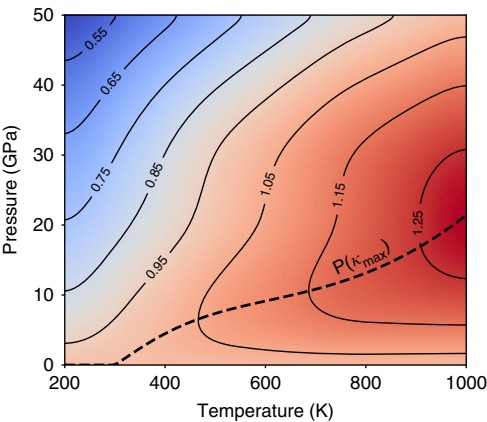

**Fig. 5** Non-monotonic pressure dependence of $\kappa^{(3+4)}$ in BSb. Pressure-temperature colormap of $\kappa^{(3+4)}$ of isotopically pure BSb scaled by the corresponding zero-pressure value at each temperature $\left(\kappa_0^{(3+4)}\right)$, similar to the plot for BAs in Fig. 4a. BSb, also being a large mass-ratio compound like BAs, is strongly affected by four-phonon scattering and shows a qualitatively similar pressure-dependent non-monotonicity of thermal conductivity like BAs

characterization of large high quality single crystals of BAs[28–30], the predictions presented here are readily accessible in experiments for BAs, and present a clear route forward following successful synthesis of BSb to realize its impressive thermal and electronic properties, predicted from first principles in this work and other studies as well[38,39]. More broadly, the approach and results presented here should also find impact in geophysical studies of heat conduction in the earth's lower mantle where temperatures can reach 4000 K and pressures can exceed 100 GPa[40–43]. One might expect that at such extreme temperatures higher-order phonon–phonon processes should play an important role in governing thermal transport. Since obtaining experimental data at lower mantle depths is challenging, the predictive approach presented here could be used to obtain important new insights into the nature of heat flow at the high temperature and pressure conditions deep in the earth.

**Table 1 Converged values of the first-principles computational parameters**

| Property | BAs | BSb | MgO | cBN |
|---|---|---|---|---|
| Kinetic energy cutoff for wave functions (Ry) | 90 | 90 | 125 | 110 |
| Kinetic energy cutoff for charge density (Ry) | 360 | 360 | 500 | 440 |
| Electronic **k**-grid for energy | $12^3$ | $6^3$-shifted | $3^3$-shifted | $4^3$-shifted |
| Electronic **k**-grid for force-displacement | $\Gamma$-shifted | $\Gamma$-shifted | $\Gamma$-shifted | $\Gamma$-shifted |
| Phonon **q**-grid for DFPT | $7^3$ | $7^3$ | $7^3$ | $7^3$ |
| No. of snapshots for anharmonic IFCs | 200 | 200 | 200 | 200 |
| Phonon **q**-grid for $\kappa$ | $17^3$ | $17^3$ | $17^3$ | $17^3$ |
| No. of nearest neighbor shells for harmonic IFCs | 11 | 11 | 11 | 11 |
| No. of nearest neighbor shells for cubic IFCs | 7 | 7 | 6 | 7 |
| No. of nearest neighbor shells for quartic IFCs | 3 | 3 | 3 | 3 |

## Methods

**Computational methodology**. The thermal conductivity tensor, $\kappa_{\alpha\beta}$, of an insulator is given by,

$$\kappa_{\alpha\beta} = \sum_\lambda C_\lambda v_{\lambda,\alpha} v_{\lambda,\beta} \tau_{\lambda,\beta}^{(\text{full})} \tag{2}$$

where the sum is over the phonon modes $\lambda \equiv (\mathbf{q}j)$ with wave vector $\mathbf{q}$ and polarization $j$, $C_\lambda$ is the volumetric heat capacity of the phonon mode $\lambda$, $v_{\lambda,\alpha}$ is the group velocity of the phonon mode $\lambda$ in the Cartesian direction $\alpha$ and $\tau_{\lambda,\beta}^{(\text{full})}$ is the transport lifetime of the phonon mode $\lambda$ obtained by iteratively solving the Peierls–Boltzmann equation (PBE) for phonon transport including three-phonon, four-phonon and phonon-isotope scattering terms, given by[28,31]:

$$\mathbf{F}_\lambda = \mathbf{F}_\lambda^0 + \tau_\lambda^{(\text{tot})} \Bigg\{ \sum_{\lambda_1\lambda_2} \Big[ W_{\lambda\lambda_1\lambda_2}^{(+)} \big(\mathbf{F}_{\lambda_2} - \mathbf{F}_{\lambda_1}\big) + \tfrac{1}{2} W_{\lambda\lambda_1\lambda_2}^{(-)} \big(\mathbf{F}_{\lambda_2} + \mathbf{F}_{\lambda_1}\big) \Big] + \sum_{\lambda_1} W_{\lambda\lambda_1}^{\text{iso}} \mathbf{F}_{\lambda_1}$$
$$+ \sum_{\lambda_1\lambda_2\lambda_3} \Big[ \tfrac{1}{6} Y_{\lambda\lambda_1\lambda_2\lambda_3}^{(1)} \big(\mathbf{F}_{\lambda_1} + \mathbf{F}_{\lambda_2} + \mathbf{F}_{\lambda_3}\big) + \tfrac{1}{2} Y_{\lambda\lambda_1\lambda_2\lambda_3}^{(2)} \big(\mathbf{F}_{\lambda_2} + \mathbf{F}_{\lambda_3} - \mathbf{F}_{\lambda_1}\big)$$
$$+ \tfrac{1}{2} Y_{\lambda\lambda_1\lambda_2\lambda_3}^{(3)} \big(\mathbf{F}_{\lambda_3} - \mathbf{F}_{\lambda_2} - \mathbf{F}_{\lambda_1}\big) \Big] \Bigg\} \tag{3}$$

Here, $1/\tau_\lambda^{(\text{tot})} = 1/\tau_\lambda^{(\text{3ph})} + 1/\tau_\lambda^{(\text{4ph})} + 1/\tau_\lambda^{(\text{iso})}$ where $1/\tau_\lambda^{(\text{3ph})} = \sum_{\lambda_1\lambda_2} \Big[ W_{\lambda\lambda_1\lambda_2}^{(+)} + \tfrac{1}{2} W_{\lambda\lambda_1\lambda_2}^{(-)} \Big]$ is the three-phonon scattering rate, $1/\tau_\lambda^{(\text{4ph})} = \sum_{\lambda_1\lambda_2\lambda_3} \Big[ \tfrac{1}{6} Y_{\lambda\lambda_1\lambda_2\lambda_3}^{(1)} + \tfrac{1}{2} Y_{\lambda\lambda_1\lambda_2\lambda_3}^{(2)} + \tfrac{1}{2} Y_{\lambda\lambda_1\lambda_2\lambda_3}^{(3)} \Big]$ is the four-phonon scattering rate and $1/\tau_\lambda^{(\text{iso})} = \sum_{\lambda_1} W_{\lambda\lambda_1}^{\text{iso}}$ is the phonon-isotope scattering rate of the phonon mode $\lambda$, $W_{\lambda\lambda_1\lambda_2}^{(\pm)}$ are the three-phonon scattering probabilities, $Y_{\lambda\lambda_1\lambda_2\lambda_3}^{(1,2,3)}$ are the four-phonon scattering probabilities, $W_{\lambda\lambda_1}^{\text{iso}}$ are the phonon-isotope scattering probabilities, and $\mathbf{F}_\lambda^0 = \hbar\omega_\lambda \mathbf{v}_\lambda \tau_\lambda^{(\text{tot})}/k_B T^2$ with $\omega_\lambda$ being the frequency and $\mathbf{v}_\lambda$ being the phonon group velocity of the phonon mode $\lambda$ and $k_B$ is the Boltzmann constant. The expressions for the three-phonon, four-phonon and phonon-isotope scattering rates are provided in the Supplementary Note 10. The transport lifetime $\left(\tau_{\lambda,\beta}^{(\text{full})}\right)$ is obtained from $\mathbf{F}_\lambda$ as $\tau_{\lambda,\beta}^{(\text{full})} = k_B T^2 F_{\lambda,\beta}/(\hbar\omega_\lambda v_{\lambda,\beta})$. It is worth noting that by solving the full PBE including three-phonon, four-phonon, and phonon-isotope scattering, we properly account for the difference between momentum-conserving Normal and resistive Umklapp scattering processes both at the three-phonon and the four-phonon level[1,2].

Using this procedure, we calculated from first principles the $\kappa^{(3)}$ and $\kappa^{(3+4)}$ of BAs and BSb as a function of temperature and pressure, and the $\kappa^{(3)}$ and $\kappa^{(3+4)}$ of MgO and cBN as a function of pressure at room temperature, for this study. The solution of the PBE requires the harmonic (second-order) and anharmonic (third and fourth-order) inter-atomic force constants (IFCs) as inputs to determine phonon modes and phonon–phonon scattering rates. We determined the harmonic IFCs under the framework of density functional perturbation theory (DFPT) implemented in the Quantum ESPRESSO (QE) package[44]. We obtained the anharmonic IFCs using the thermal snapshot IFC fitting technique developed in ref. [28,31], which also accounts for the effects of the temperature-dependent atomic displacements, the zero-point motion of atoms and the polar effects of the optic phonons on the anharmonic IFCs as well. In this thermal snapshot IFC fitting technique, the harmonic forces are subtracted out from the force-displacement dataset using the short and long-range harmonic IFCs from QE-DFPT, and only the anharmonic IFCs are fit to the remaining forces [for additional details, see Appendix A of ref. [31]]. We obtained the required force-displacement dataset for the thermal snapshot IFC fitting technique under the framework of the density functional theory as implemented in QE.

We also included the renormalization of the harmonic and anharmonic IFCs developed in our prior work[31] to ensure that the phonon quasiparticles are well-defined, particularly at high temperatures, where the anharmonicity of materials could be strong. The renormalized harmonic IFCs ($\Psi_{\alpha\beta}(N\nu, P\pi)$) are obtained from the bare (unrenormalized) IFCs ($\Phi_{\alpha\beta}(N\nu, P\pi)$ and $\Phi_{\alpha\beta\gamma\delta}(N\nu, P\pi, Q\eta, R\rho)$) through the following equation[31]:

$$\Psi_{\alpha\beta}(N\nu, P\pi) = \Phi_{\alpha\beta}(N\nu, P\pi) + \frac{\hbar}{4N_0} \sum_{QR} \sum_{\eta\rho} \sum_{\gamma\delta} \sum_{\mathbf{q}s} \Phi_{\alpha\beta\gamma\delta}(N\nu, P\pi, Q\eta, R\rho)$$
$$\times \frac{W_\gamma(\eta;\mathbf{q}s)W_\delta^*(\rho;\mathbf{q}s)}{\Omega_{\mathbf{q}s}\sqrt{M_\eta M_\rho}} e^{i\mathbf{q}\cdot(\mathbf{R}(Q)-\mathbf{R}(R))} \big(2n_{\mathbf{q}s} + 1\big) \tag{4}$$

Here, the renormalized phonon frequencies ($\Omega_{\mathbf{q}s}$) and the renormalized eigenvectors ($\mathbf{W}(\nu, \mathbf{q}s)$) depend on the renormalized harmonic IFCs ($\Psi_{\alpha\beta}(N\nu, P\pi)$). Therefore, Eq. (4) has to be solved iteratively in a self-consistent manner. Note that Eq. (4) does not contain the bare cubic IFCs ($\Phi_{\alpha\beta\gamma}(N\nu, P\pi, Q\eta)$) [see Appendix C in ref. [31] for the derivation of Eq. (4)]. Once $\Psi_{\alpha\beta}(N\nu, P\pi)$ are obtained, the renormalized anharmonic IFCs ($\Psi_{\alpha\beta\gamma}(N\nu, P\pi, Q\eta)$ and $\Psi_{\alpha\beta\gamma\delta}(N\nu, P\pi, Q\eta, R\rho)$) are obtained by refitting the original force-displacement data using $\Psi_{\alpha\beta}(N\nu, P\pi)$ as input [see Supplementary Note 8 and 9 for comparisons between renormalized and bare/unrenormalized calculations]. We obtained the pressure in our calculations by taking the derivative of the fourth-order anharmonic Helmholtz free energy ($F_{4\text{th}-\text{order}}$) with respect to volume ($V$) at each temperature ($T$) as $P(a) = -\frac{\partial F_{4\text{th}-\text{order}}}{\partial V}\big|_{T,a} \approx -\frac{F_{4\text{th}}(a+\Delta a)-F_{4\text{th}}(a-\Delta a)}{V(a+\Delta a)-V(a-\Delta a)}\big|_T$, where $a$ is the lattice constant and $\Delta a \sim 0.05\%$ of $a$. The expression for the fourth-order anharmonic Helmholtz free energy is given in the Supplementary Note 10.

**Convergence parameters**. We used norm-conserving pseudopotentials under the local density approximation for all the elements in our calculations in this manuscript [see Supplementary Note 3 for the temperature and pressure-dependent $\kappa$-calculations for BAs using PBEsol pseudopotentials]. We systematically checked for convergence of various computational parameters used in our first-principles framework, and the converged values of the parameters are listed in Table 1.

For the force-displacement calculations, we used thermal snapshots on $5 \times 5 \times 5$ supercells of atoms (250 atoms per supercell). Using the computational parameters listed in Table 1, the total energy converged to less than $9 \times 10^{-4}$ Ry per unit cell, the total stress converged to less than 0.8 kbar per unit cell and the forces on atoms converged to less than $10^{-5}$ Ry/au. The phonon frequencies and eigenvectors were unchanged by adding one more shell of harmonic IFCs. All reported thermal conductivity values are converged with respect to the number of snapshots, number of nearest neighbors of the anharmonic (cubic and quartic) IFCs and the **q**-grid for all materials in this work. For example, Table 2 shows the convergence of the thermal conductivity of isotopically pure BAs with respect to these parameters at four different pressure-temperature conditions.

**Computational challenges and code efficiency to reduce cost**. To mitigate the computational cost of obtaining the anharmonic IFCs and solving the 3 + 4-phonon PBE on a pressure-temperature grid, we have developed several computational efficiencies and leveraged the advances in the hardware architecture of the computer processors, without comprising accuracy. Notably,

1. To obtain the cubic and quartic IFCs in a computationally efficient manner, we have developed a thermal snapshot technique in our earlier work[31] which requires force-displacement calculations using only about 200 snapshots of thermally relevant configurations of atoms in the supercell for each point on the pressure-temperature grid. Furthermore, our thermal snapshot technique seamlessly includes the effects of the zero-point motion of atoms and polar

**Table 2 Convergence of the thermal conductivity of isotopically pure BAs**

| Property | 300 K, 0 GPa | 300 K, 17 GPa | 1000 K, 0 GPa | 1000 K, 76 GPa |
|---|---|---|---|---|
| Parameters in Table 1 | 1331 (3102) | 1486 (2356) | 154 (891) | 186 (279) |
| 150 snapshots | 1329 (3107) | 1489 (2342) | 155 (895) | 185 (279) |
| 1 more cubic and quartic IFC shell | 1349 (3183) | 1460 (2335) | 153 (923) | 185 (283) |
| $21^3$ **q**-grid for $\kappa$ | 1333 (3099) | 1491 (2360) | 155 (891) | 190 (284) |

Convergence of $\kappa^{(3)}$ and $\kappa^{(3+4)}$ of isotopically pure BAs with respect to the number of snapshots and number of nearest neighbors of the anharmonic (cubic and quartic) IFCs and the **q**-grid. From the second row onwards, the property column indicates the change in the computational parameters from those listed in Table 1. The numbers outside parentheses are $\kappa^{(3+4)}$ [W m$^{-1}$ K$^{-1}$], while the numbers within parentheses are $\kappa^{(3)}$ [W m$^{-1}$ K$^{-1}$]

effects (splitting between the longitudinal and transverse optic phonons at the Γ-point) on the anharmonic IFCs, and also the temperature dependence of the anharmonic IFCs, without any ad-hoc adjustments to the formulation. In contrast, the widely-used technique that holds most atoms in the supercell at their classical equilibrium positions[27] cannot capture these effects, and requires force-displacement calculations on several hundreds to a few thousands of supercells for each point on the pressure-temperature grid, particularly for the quartic IFCs, making it prohibitively expensive for studies of the temperature and pressure dependence of four-phonon scattering in semiconductors.

2. As described in our earlier work[31], to solve the PBE (Eq. 3) on a grid of **q**-points in the Brillouin zone, the main challenge is to efficiently compute and store the four-phonon matrix elements $\left(\Psi_{\lambda\lambda_1\lambda_2\lambda_3}\right)$. For example, for each point on the pressure-temperature grid, and for a $17^3$ Brillouin zone grid (**q**-grid) used in this study, there are about 175 million three-phonon matrix elements $\left(\Psi_{\lambda\lambda_1\lambda_2}\right)$ in the irreducible Brillouin zone for the phonon mode $\lambda \sim$ (**q**, $j$), while there are about 5.2 trillion four-phonon matrix elements $\left(\Psi_{\lambda\lambda_1\lambda_2\lambda_3}\right)$, a factor of thirty-thousand larger (all materials in this study have two atoms in each unit cell, thus have 6 phonon polarizations for each **q**-point). Furthermore, since Eq. (3) is solved iteratively in this study, the four-phonon matrix elements $\left(\Psi_{\lambda\lambda_1\lambda_2\lambda_3}\right)$ must not only be computed, but also stored in files for use in subsequent iterations. Fortunately, since the analytical tetrahedron scheme used to calculate the energy-conserving $\delta$-functions in the expression for the scattering rates (see Supplementary Note 10) provides higher accuracy than the widely-used adaptive Gaussian smearing scheme (see e.g., ref. [22]) or the non-adaptive Lorentzian scheme[27], a $17^3$ **q**-grid has been sufficient for the convergence of $\kappa^{(3)}$ and $\kappa^{(3+4)}$ for the materials in this study, as described in the Methods section. Furthermore, we reduce the cost of computing the three-phonon matrix elements by 50% and the four-phonon matrix elements by 83% by invoking the following transposition symmetries of the matrix elements: $\Psi_{\lambda\lambda_1\lambda_2} = \Psi_{\lambda\lambda_2\lambda_1}$ and $\Psi_{\lambda\lambda_1\lambda_2\lambda_3} = \Psi_{\lambda\lambda_1\lambda_3\lambda_2} = \Psi_{\lambda\lambda_2\lambda_1\lambda_3} = \Psi_{\lambda\lambda_2\lambda_3\lambda_1} = \Psi_{\lambda\lambda_3\lambda_1\lambda_2} = \Psi_{\lambda\lambda_3\lambda_2\lambda_1}$ [45] (We do not invoke the transposition symmetries on the first phonon mode $\lambda$ in these matrix elements, since $\lambda$ is restricted to the irreducible **q**-grid). Finally, we also mitigate the computational load by eliminating three-phonon and four-phonon processes that have vanishing phase space (i.e., the energy conserving $\delta$-functions in the expressions for the scattering rates in Supplementary Note 10) beforehand, and improve the performance of our code by leveraging vectorized computation and optimized cache memory usage.

**Code availability**. All formulations and algorithms necessary to reproduce the results of this study are described in the Methods section, in the Supplementary Information and in ref. [31].

## Data availability

The data supporting the findings of this work are available from the corresponding author upon reasonable request.

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

## Acknowledgements

This work was supported by the Office of Naval Research under a MURI, Grant No. N00014-16-1-2436. We acknowledge the National Energy Research Scientific Computing Center (NERSC), a U.S. Department of Energy Office of Science User Facility operated under Contract No. DE-AC02-05CH11231, the Extreme Science and Engineering Discovery Environment (XSEDE), which is supported by National Science Foundation grant number ACI-1548562, and the Boston College Linux clusters for the computational resources and support.

## Author contributions

N.K.R and D.B. originated the research. N.K.R. performed the ab initio calculations. N.K.R. and D.B. analyzed the results and wrote the manuscript. Both authors studied, commented on, and edited the manuscript.

## Additional information

**Competing interests:** The authors declare no competing interests.

