## [Peer Review File · Nature Communications]

Reviewers' comments:

Reviewer #1 (Remarks to the Author):

The authors have developed and deployed a first-principles methodology to study the bulk thermal conductivity of BAs, a high thermal conductivity material that has drawn much interest recently, culminating in successful synthesis of large crystals of BAs. It was also found that 4-phonon processes play a significant role in this material's thermal conductivity. The authors study the pressure dependence of 3 and 4-phonon processes in BAs and find that they combine into a non-monotonic trend of thermal conductivity vs pressure. This is a challenging and landmark study, which I strongly support for publication after minor questions/concerns are addressed.

1. why is the fit to MgO Fig. 1b so poor despite the whole renormalization machinery?
2. How are the 4-phonon coupling constants computed? The authors point the reader to their earlier work, but then in that work I had to dig down to appendix A and B for details. Is a 5x5x5 supercell still used in computing the 4th order IFCs and how do the authors ensure it is sufficiently large?
3. Since the non-monotonic behavior is clearly temperature dependent (more monotonic at low T and less at high T), why not plot the 3-phonon and 4-phonon scattering rates as a function of T? It is thought, based on simple empirical models, that the 4-phonon processes have a stronger T dependence and they do show up more in high T portions of most of the curves in this paper. Figure 3 looks at frequency-dependent trends rather than temperature dependence, which is useful but somewhat at odds with how Figure 2 is presented.

Minor:

1. typo in figure 2 caption, subfigure c appears to be also called b.

Reviewer #2 (Remarks to the Author):

The work by Ravichandran and Broido addresses the effect of pressure on the thermal conductivity of BAs, an emergent material with extremely high thermal conductivity.

The authors use a state-of-the-art ab initio approach based on the Boltzmann Peierls transport equation for phonons. The original BP approach has been extended by these same authors to treat more accurately the temperature dependence of the thermal conductivity in systems that undergo significant thermal expansion and where the effect of 4th order anharmonic phonon-phonon scattering needs to be taken into account. In this work they show that using this approach is essential to capture the correct qualitative feature of non-monotonic pressure dependence of the thermal conductivity of BAs. In addition, they also extend their study to BSb.

The effect showcased in this paper is novel and the authors propose a sufficiently rich analysis that digs into the physical origin of the calculated results.

The paper reads well and the figures are clear exhaustive.

Calculations are described in detail, and the authors report convergence parameters, which appear quite convincing, however I would like to have a few clarification about how the calculations are performed.

- 1) As for convergence it is not clear whether the number of snapshots used for the calculation of the anharmonic IFC is sufficient, as the effect of this parameter is not clearly explained.
- 2) Also, to compute the anharmonic IFC a supercell approach is needed, and I cannot find mention of the size of such supercell (number of replicas). From my experience, this is a rather critical issue, when force constants are long range.

- 3) MgO is an ionic material: is the contribution of the non-analytical term (Born effective charges) considered in the calculation of the phonons? There is probably no need for it in BAs.
- 4) How sensitive are these results to the choice of the exchange and correlation functional? LDA may not be the best choice, and it would be useful to report a comparison with a GGA functional (PBEsol was used for NaCl in ref. 31).

In conclusion, I recommend that this work is published in Nature Commun. provided that the author address the issues reported below.

Reviewer #3 (Remarks to the Author):

Overall I found this to be an excellent paper and strongly recommend it for publication. The anomalous behavior identified is fascinating and was studied with sufficient detail that it is very convincing. The paper was well written and the work was thorough and well described. There are two suggestions I have for possible improvement of the paper, which I believe will enable readers to obtain the main insights more quickly:

1. Is it possible to generate a figure that shows k (3phonon only) vs. P , k (4phonon only) vs P , and then $k(3 + 4 \text{ phonon})$ vs. P ? All of the figures show only the first and last plots of the series I've listed, but the k (4phonon only) is arguably the most important. The 3 phonon plots are generally monotonic, but the 4phonon only plots should be the ones that illustrate where the non-monotonic behavior originates. Also, my suggestion would be to show less plots of the scattering rates – (still show a few), but instead, show the scattering phase space, since that's a single number at each pressure for a given type of interaction. That way you can clearly show that the knob being tuned is mostly the scattering phase space and it will correspondingly show the features that show up in k .
2. There was a key sentence in the SI, "The expressions for the three-phonon, four-phonon and phonon-isotope scattering probabilities, the computational challenges associated with these calculations and our strategies to overcome these challenges are discussed in our prior work [31]". This sentence or a few sentences pointing this out need to be in the body of the paper early. The authors should give the readers context early on that these types of calculations are not standard and have only recently become possible. Although the authors cite ref 31, it would be useful if they added a brief validation section at the beginning of the SI, demonstrating to the reader that their 3+4 phonon code works and reproduces experiments correctly. Since the addition of 4 phonon scattering is the heart of this paper, more deference to the level of investment required to perform such calculations should be given. Otherwise, readers from outside fields may come away with a misconception that such calculations are easy/straightforward and routine.

Lastly, I think it would also be useful for the authors to comment on how this effect might be useful from a technological standpoint. As presented, this is purely a basic physics discovery. However, the impact of the paper might be higher if the authors were to provide some vision for what this could possibly be useful for. Such conjecture could be the key to enticing experimentalists to go and check if this prediction is correct or not. Otherwise, if it is only an anecdotal physical effect, it may remain an unchecked prediction. In this sense, I do not think the authors need to do a lot, but a few comments on possibilities is useful to help others determine what else might be possible. The experimental studies of Bas thermal conductivity were motivated by the possibility of competing with diamond. As a result, experimentalists chased the possibility and eventually confirmed it. What would be the motivation here?

Please also note the typo in the Fig. 2 caption, it does not mention panel c, but instead b twice - which is incorrect.

Response to the reviewers' comments for the manuscript titled "Non-monotonic Pressure Dependence of the Thermal Conductivity of Boron Arsenide" by Navaneetha K. Ravichandran and David Broido submitted to Nature Communications.

We thank the reviewers for taking the time to carefully read the manuscript and for the useful comments. All three reviewers have recommended publication of this manuscript, provided we address their points. We have responded to all of the reviewers' comments and suggestions in the response/rebuttal report below. Our responses are in blue.

Reviewer #1:

1. The reviewer asks, "Why is the fit to MgO Fig. 1b so poor despite the whole renormalization machinery?"

To address the reviewer's question, we have added a discussion in the Supplementary Note 9 of the revised supplementary material. The renormalization procedure has a weak effect on the thermal conductivity of MgO, as shown in fig. S17 (a) of the revised supplementary material, and so cannot explain the observed discrepancy with the experiments from ref. [9] in fig. 1(b) of the main manuscript. The observed discrepancy could be attributed to the presence of impurities in the sample used in ref. [9]. For example, addition of a small amount (500 parts per million [ppm]) of Fe impurities, one of the known impurities in MgO, as mass defects on the Mg site to our calculation improves the qualitative and quantitative agreement with the experiments, as shown below and in fig. S17 (b) of the revised supplementary material. Other sources of error from measurement uncertainties, such as the pressure dependence of the heat capacities for the metallic transducer used in the time-domain thermoreflectance (TDTR) experiment in ref. [9] and that of MgO at high pressure, which are used as input to obtain the thermal conductivity of MgO in the TDTR experiment, may also exist.

2. The reviewer asks, “How are the 4-phonon coupling constants computed? The authors point the reader to their earlier work, but then in that work I had to dig down to appendix A and B for details. Is a 5x5x5 supercell still used in computing the 4th order IFCs and how do the authors ensure it is sufficiently large?”

To address the reviewer’s question, we have added more details on how we compute the harmonic and anharmonic interatomic force constants (IFCs), and the three-phonon, four-phonon and phonon-isotope matrix elements to the Methods section in the revised manuscript. We have indeed used a 5X5X5 supercell to compute the cubic and the quartic IFCs using the supercell thermal snapshot approach, while used a 7X7X7 q-grid to calculate the harmonic IFCs using density functional perturbation theory. As described in the Methods subsection B of the revised manuscript, we included 11 nearest neighbors for the harmonic IFCs, 7 nearest neighbors for the cubic IFCs and 3 nearest neighbors for the quartic IFCs for BAs. Including one more nearest neighbor for cubic and quartic IFCs produced only small changes to the calculated thermal conductivity (see table II of the Methods subsection B in the revised manuscript). For example, the calculated BAs thermal conductivity at room temperature and zero pressure changed from 1331 W/m-K to 1349 W/m-K, a difference of only about 1%. Addition of one more nearest neighbor for harmonic IFCs left the phonon frequencies and eigenvectors unchanged. The inter-atomic distances involved in the converged cubic and quartic IFC tensors are less than 30% and 20% of the total size of the supercell respectively; thus, a supercell larger than 5X5X5 repeated unit cells is not required for the force-displacement calculations.

3. The reviewer suggests, “Since the non-monotonic behavior is clearly temperature dependent (more monotonic at low T and less at high T), why not plot the 3-phonon and 4-phonon scattering rates as a function of T ? It is thought, based on simple empirical models, that the 4-phonon processes have a stronger T dependence and they do show up more in high T portions of most of the curves in this paper. Figure 3 looks at frequency-dependent trends rather than temperature dependence, which is useful but somewhat at odds with how Figure 2 is presented.”

We thank the reviewer for this suggestion. We have now included temperature-dependent plots of three-phonon and four-phonon scattering rates at different pressures in the Supplementary Note 6 of the revised supplementary material, similar to the plots in fig. 3(b) of ref. [27]. We find that, indeed, the four-phonon scattering rates have stronger temperature dependence than three-phonon processes at 0 GPa and 50 GPa, consistent with the previous findings at ambient pressure [27]. This stronger temperature dependence of four-phonon scattering is the root cause of the temperature dependence of the non-monotonic behavior of $k^{(3+4)}$ with pressure rise, as observed in the main text.

We note that fig. 4 (b) of this and the earlier version of the manuscript present a picture of the small frequency range within which the three-phonon and four-phonon scattering rates at 300 K and 750 K overlap at different pressures. This allows comparison of their opposing responses to pressure rise and highlights the role of temperature in further tuning the competition between the two processes as the pressure is increased. Therefore, we have retained those plots in the revised manuscript as well.

4. We thank the reviewer for pointing out “the typo in figure 2 caption, subfigure c appears to be also called b”. We have fixed the typo.

Reviewer #2:

1. The reviewer says, *“As for convergence it is not clear whether the number of snapshots used for the calculation of the anharmonic IFC is sufficient, as the effect of this parameter is not clearly explained.”*

To address the reviewer’s comment, we have now included the convergence of our calculations with respect to the number of snapshots in the Methods subsection B of the revised manuscript (see new table II). We find that the 3+4-phonon limited thermal conductivity calculations, originally performed with 200 snapshots, are sufficiently converged from low to high temperatures, and from low to high pressures. For example, taking 150 snapshots instead of 200 snapshots only changes the zero pressure BAs thermal conductivity at 300 K from 1331 W/m-K to 1329 W/m-K.

2. The reviewer says, *“Also, to compute the anharmonic IFC a supercell approach is needed, and I cannot find mention of the size of such supercell (number of replicas). From my experience, this is a rather critical issue, when force constants are long range.”*

To address the reviewer’s question, we have now included this convergence information in the Methods subsection B of the revised manuscript (please see response to Reviewer #1, point 2 above). Our tests show that the quantities presented in the original manuscript are converged.

3. The reviewer asks, *“MgO is an ionic material: is the contribution of the non-analytical term (Born effective charges) considered in the calculation of the phonons? There is probably no need for it in BAs.”*

Yes, the non-analytical terms have been included to calculate phonon dispersions for all materials considered in our manuscript (as shown by the splitting observed between the transverse and longitudinal optic phonon branches at the Gamma point in the phonon dispersion plots, particularly evident in fig. 1 (a) of the main text for MgO). The non-analytical terms are also considered in the calculation of the cubic and quartic IFCs through the displacements of atoms in the thermal snapshots, and through the anharmonic IFC fitting procedure by subtracting the short and long-range harmonic contributions to the total forces, as described in our prior work (Ref. [31]). We have included this information in the Methods subsection A of the revised manuscript.

4. The reviewer asks, *“How sensitive are these results to the choice of the exchange and correlation functional? LDA may not be the best choice, and it would be useful to report a comparison with a GGA functional (PBEsol was used for NaCl in ref. 31).”*

We thank the reviewer for this suggestion. We have now included a separate Supplementary Note 3 in the supplementary material, showing the pressure and temperature-dependent thermal conductivity results for BAs using an ultrasoft pseudopotential with PBEsol (GGA type) exchange correlation functional. We find that the main novel features of (1) non-monotonic pressure-dependence of the 3+4-phonon thermal conductivity of BAs and the resulting thermal conductivity peak, (2) the temperature dependence of the pressure at which the thermal conductivity achieves a maximum, and (3) the larger percentage enhancement of

thermal conductivity at its peak compared to the zero pressure value at elevated temperatures, are also observed in the PBEsol calculations. This is seen in fig. S6 (c) of the revised supplementary material and also included below. There are some quantitative differences between the two calculations caused by slightly larger lattice constants, lower group velocities, smaller three-phonon scattering rates and larger four-phonon scattering rates in PBEsol compared to the LDA results (fig. S8 of the revised supplementary material), which are consistent with that reported in the literature for the thermal and thermodynamic properties of BAs and other materials using these two types of pseudopotentials [10-13].

Reviewer #3:

1. The reviewer asks, “Is it possible to generate a figure that shows k (3phonon only) vs. P , k (4phonon only) vs. P , and then $k(3 + 4\text{ phonon})$ vs. P ? All of the figures show only the first and last plots of the series I’ve listed, but the k (4phonon only) is arguably the most important. The 3 phonon plots are generally monotonic, but the 4phonon only plots should be the ones that illustrate where the non-monotonic behavior originates. Also, my suggestion would be to show less plots of the scattering rates – (still show a few), but instead, show the scattering phase space, since that’s a single number at each pressure for a given type of interaction. That way you can clearly show that the knob being tuned is mostly the scattering phase space and it will correspondingly show the features that show up in k .”

To address the reviewer’s comment, we have now included a comparison of the thermal conductivity including three-phonon processes only, four-phonon processes only, and both three-phonon and four-phonon processes for BAs in the Supplementary Note 4 of the revised supplementary material. Though inclusion of four-phonon scattering strongly affects the thermal conductivity of BAs, this effect is not due to particularly strong four-phonon scattering; rather it is due to the unusually weak three-phonon scattering in BAs in a narrow frequency range where the four-phonon scattering is at its largest, as seen in fig. 3 of the main manuscript (included below for zero pressure and room temperature). Away from this frequency range (~4-8 THz), three-phonon scattering rates are much larger than the four-phonon scattering rates in BAs. Thus, the thermal conductivity including only four-phonon scattering [$k^{(4)}$] is much larger than that including only three-phonon scattering [$k^{(3)}$] from

below room temperature up to 500K. For example, at 300K, $k^{(4)} = 15332 \text{ W/m-K}$, about five times larger than $k^{(3)}$. Since the four-phonon scattering strength increases more rapidly with temperature than three-phonon scattering, $k^{(4)}$ starts to become comparable to $k^{(3)}$ at the highest temperatures. Thus, at 1000K, $k^{(4)}$ is only about 10% larger than $k^{(3)}$. Note that the four-phonon scattering rates in BAs are still orders of magnitude smaller than those of strongly anharmonic materials like NaCl or PbTe [see, e.g. fig. S3 (c) in Ref. 31, or fig. 2 (b) in: Yi Xia, Appl. Phys. Lett. 113, 073901, 2018].

We have also included plots of three-phonon and four-phonon scattering phase spaces of the acoustic modes of BAs as functions of pressure at different temperatures in the Supplementary Note 5 of the revised supplementary material. As seen from fig. S11 of the revised supplementary material (also included below), the dominant three-phonon scattering channels are the AAA processes (scattering between three acoustic phonons). AAO and AOO scattering phase spaces are small. The four-phonon scattering channels with the largest phase space are the AAOO processes.

The three-phonon and four-phonon phase spaces show opposite responses to increasing pressure. The AAA phase space increases monotonically with increasing pressure, qualitatively consistent with the corresponding increase in three-phonon scattering rates shown in fig. 3 of the main text. The AAOO phase space decreases monotonically with increasing pressure, also qualitatively consistent with the corresponding weakening of the AAOO scattering rates.

However, the phase spaces present only a part of the overall picture for the non-monotonic pressure dependence of $k^{(3+4)}$ of BAs. For example, using only the phase space information, one might *erroneously* conclude that:

(i) *Four-phonon scattering (AAOO scattering in particular) dominates at all pressures since its phase space is much larger than the three-phonon phase space;* if that were true, the BAs thermal conductivity would be dominated by four-phonon scattering only and, therefore, increase monotonically with pressure, which is contrary to the non-monotonic behavior found from the full calculations.

(ii) *AOO scattering is important at zero pressure, since the AOO phase space is only three times smaller than the AAA phase space.* In fact, AOO scattering rates are negligibly small.

(iii) *AAAA scattering is always weaker than AAOO scattering.* In fact, it dominates at high pressures (see e.g. fig. 3 (d) in the main text).

The reason for the above erroneous conclusions from considering only the phase space is that the total scattering rates, which eventually determine $k^{(3+4)}$, have several factors in addition to the phase space (e.g. scattering matrix elements and Bose factors), which can have different pressure and temperature dependencies compared to the phase space. Additionally, the competition between three-phonon and four-phonon scattering processes vs. temperature cannot be captured from the respective phase spaces alone, since they show very weak temperature dependence in fig. S11 of the revised supplementary material (also shown above).

Finally, as noted earlier, the competition between three-phonon and four-phonon processes occurs within a small region of the Brillouin zone, while $k^{(4)}$ and the phase spaces are averages over the entire Brillouin zone. Thus, we believe that the frequency-dependent plots of three-phonon and four-phonon scattering rates present a more complete description of the non-monotonic pressure dependence of $k^{(3+4)}$ for BAs, and so, we have retained them in the revised manuscript as well.

2. The reviewer suggests, *“There was a key sentence in the SI, “The expressions for the three-phonon, four-phonon and phonon-isotope scattering probabilities, the computational challenges associated with these calculations and our strategies to overcome these challenges are discussed in our prior work [31]”. This sentence or a few sentences pointing this out need to be in the body of the paper early. The authors should give the readers context early on that these types of calculations are not standard and have only recently become possible. Although the authors cite ref 31, it would be useful if they added a brief validation section at the beginning of the SI, demonstrating to the reader that their 3+4 phonon code works and reproduces experiments correctly. Since the addition of 4 phonon scattering is the heart of this paper, more deference to the level of investment required to perform such calculations should be given. Otherwise, readers from outside fields may come away with a misconception that such calculations are easy/straightforward and routine.”*

We thank the reviewer for this suggestion. We have now included these points explicitly in the revised manuscript. Our previous works on BAs [28], and Sodium Chloride and diamond [31] show that the unified first principles framework used in the current study is able to reproduce the experimental phonon dispersions, thermal expansion and thermal conductivity over a wide temperature range for both weakly anharmonic and strongly anharmonic materials. Also, from fig. 1(b) of the main text and fig. S17 of the revised supplementary material, our approach is also able to capture the experimental pressure dependence of the thermal conductivity of MgO at room temperature reasonably well. Furthermore, we have also shown in the Supplementary Note 1 of the revised supplementary material that our approach is able to predict the change in transverse acoustic phonon frequency at the Gamma-point as a function of volume and the pressure-volume curve for cubic Boron Nitride, and the pressure-volume curve for MgO at room temperature.

To emphasize the accuracy of our approach, we have added the following statements to page 3 of the revised main text - *“...To calculate κ of a solid as a function of pressure and temperature, we have implemented a predictive, first principles approach with no adjustable parameters, that has demonstrated good agreement with the measured lattice expansion, temperature-dependent phonon frequencies and thermal conductivity of diamond and Sodium Chloride [31], and the ultra-high thermal conductivity of BAs [28], where both three-phonon and four-phonon scattering are important. This recently developed approach goes beyond the standard calculations by including higher-order four-phonon scattering processes along with three-phonon processes in describing phonon transport, thereby providing a new opportunity to investigate the significance of higher-order processes and their interplay with those of lower order on the thermal properties of materials...”*

To highlight the computational challenges with these calculations and the code efficiencies that we have developed to overcome them, we have included a separate subsection C in the Methods section and also added the following statement in page 3 of the revised manuscript, *“...These calculations present major computational challenges, and we have developed several efficiencies in our code to overcome them, as described in the Methods section C...”*

3. The reviewer suggests, *“Lastly, I think it would also be useful for the authors to comment on how this effect might be useful from a technological standpoint. As presented, this is purely a basic physics discovery. However, the impact of the paper might be higher if the authors were to provide some vision for what this could possibly be useful for. Such conjecture could be the key to enticing experimentalists to go and check if this prediction is correct or not. Otherwise, if it is only an anecdotal physical effect, it may remain an unchecked prediction. In this sense, I do not think the authors need to do a lot, but a few comments on possibilities is useful to help others determine what else might be possible. The experimental studies of Bas thermal conductivity were motivated by the possibility of competing with diamond. As a result, experimentalists chased the possibility and eventually confirmed it. What would be the motivation here?”*

We thank the reviewer for this suggestion. We have now included the following discussion in the summary of the manuscript that highlights the potential impact and technological importance of our findings from this study: *“...Apart from the ultra-high thermal conductivity of BAs at ambient conditions, existing literature has also predicted that BAs possesses excellent electron and hole mobilities as well [39], and that the application of hydrostatic pressure could*

enhance its electronic properties further [40]. These studies have also predicted enhanced carrier mobilities in BSb at ambient conditions and under pressure, for the same reasons as BAs. Our results reveal the novel way in which pressure tunes the ultra-high thermal conductivity of BAs and BSb, and also emphasize the importance of including higher-order phonon processes to explore novel heat transfer regimes that upend conventional understanding. With the recent advances in the growth and characterization of large high quality single crystals of BAs [28-30], the predictions presented here are readily accessible in experiments for BAs, and present a clear route forward following successful synthesis of BSb to realize its impressive thermal and electronic properties, predicted from first principles in this work and other studies as well [39, 40]. More broadly, the approach and results presented here should also find impact in geophysical studies of heat conduction in the earth's lower mantle where temperatures can reach 4000 K and pressures can exceed 100 GPa [41-44]. One might expect that at such extreme temperatures high-order phonon-phonon processes should play an important role in governing thermal transport. Since obtaining experimental data at lower mantle depths is challenging, the predictive approach presented here could be used to obtain important new insights into the nature of heat flow at the high temperature and pressure conditions deep in the earth."

4. We thank the reviewer for pointing out, "Please also note the typo in the Fig. 2 caption, it does not mention panel c, but instead b twice - which is incorrect." We have fixed this typo.

REVIEWERS' COMMENTS:

Reviewer #1 (Remarks to the Author):

The authors have addressed all my comments and concerns. Their review is thorough.

Reviewer #2 (Remarks to the Author):

The authors have very carefully addressed all the reviewers concerns.

The manuscript is original and of very good quality, and all the calculations are proven rigorous.

I recommend its publication.